

# CDMFT+HFD: An extension of dynamical mean field theory for nonlocal interactions applied to the single band extended Hubbard model

**Sarbajaya Kundu[1⋆] and David Sénéchal[2]**

**1** Department of Physics, University of Florida, Gainesville, FL 32611, USA
**2** Département de physique and Institut quantique, Université de Sherbrooke, Sherbrooke, Québec, Canada J1K 2R1

⋆ sarbajay.kundu@ufl.edu

## Abstract

We examine the phase diagram of the extended Hubbard model on a square lattice, for both attractive and repulsive nearest-neighbor interactions, using CDMFT+HFD, a combination of Cluster Dynamical Mean Field theory (CDMFT) and a Hartree-Fock mean-field decoupling of the inter-cluster extended interaction. For attractive non-local interactions, this model exhibits a region of phase separation near half-filling, in the vicinity of which we find islands of $d$-wave superconductivity, decaying rapidly as a function of doping, with disconnected regions of extended $s$-wave order at smaller (higher) electron densities. On the other hand, when the extended interaction is repulsive, a Mott insulating state at half-filling is destabilized by hole doping, in the strong-coupling limit, in favor of $d$-wave superconductivity. At the particle-hole invariant chemical potential, we find a first-order phase transition from antiferromagnetism (AF) to $d$-wave superconductivity as a function of the attractive nearest-neighbor interaction, along with a deviation of the density from the half-filled limit. A repulsive extended interaction instead favors charge-density wave (CDW) order at half-filling.

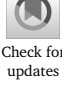

# 1   Introduction

The single-band Hubbard model has long served as a useful platform for studying the effect of strong electronic correlations [1–6]. In particular, it explains many of the experimental observations in the high-$T_c$ cuprate superconductors [2,7–16], providing an approximate picture for the description of these materials [17–25]. More recently, there have been numerous studies on extensions of this model with nearest-neighbor interactions, known as the extended Hubbard model (EHM) [26–90]. There are several reasons for the continuing interest of the community in exploring the effect of non-local interactions. In actual materials, the interactions between neighboring sites may not be completely screened, necessitating a more careful treatment of longer-range interactions. The model with an attractive nearest-neighbor interaction provides an effective representation of the attractive interactions mediated by electron-phonon coupling, and may be realized in ultra-cold atom systems. The relevance of studying such a model is further emphasized by recent ARPES studies on the one-dimensional cuprate chain compound $Ba_{2-x}Sr_xCuO_{3+\delta}$ [91], where the observations can be explained using a Hubbard model with an attractive extended interaction. On the other hand, the model with repulsive non-local interactions provides an ideal playground for studying the interplay of charge and spin fluctuations, since the relative magnitude of the charge fluctuations can be controlled by the strength of the extended interaction [26,30,34,35]. The EHM at quarter-filling has proven useful for describing the charge ordering transition due to inter-site Coulomb interactions in a variety of materials [28, 48, 49, 79, 83]. Both the Hubbard model and its extension with longer-range interactions have contributed significantly to the methodological development in the field of strongly correlated systems, and in particular high-$T_c$ superconductors, which is essential for obtaining results that can be quantitatively compared with experiments.

In recent years, the properties of the EHM have been analyzed using a variety of approaches, including, among others, mean-field theory [50–52,72], functional renormalization group (fRG) [39], exact diagonalization (ED) [29,32,55,61], density-matrix renormalization group (DMRG) [57,63], Quantum Monte Carlo (QMC) [70,87,89,92] and the fluctuation-exchange approximation (FLEX) [56]. However, many of the approaches used are best suited for studying the weak-coupling or the strong-coupling limit, and there are few that can describe the intermediate-coupling regime equally well. Even among those that can, each has its own limitations. For instance, simple exact diagonalizations are restricted to small systems, quantum Monte Carlo methods suffer from the fermion sign problem in many applications of interest, the density-matrix renormalization group (DMRG) applies to one-dimensional or ribbon-like systems, etc. In addition, certain aspects of the model with repulsive interactions have been studied in detail using the so-called extended dynamical mean-field theory (EDMFT) approach [93–95], in which the local density fluctuations together with the local self-energy are propagated on the whole lattice using the known dispersion and density-density extended interactions. Other variations of this method, such as a combination of EDMFT with the GW approximation [27, 96–98], which perturbatively includes non-local self-energy corrections, and the dual boson method [81, 82, 99], which constructs a diagrammatic expansion about

the extended DMFT, have likewise contributed to its understanding. More recently, cluster methods [26, 38, 76–78, 100, 101], which capture short-range correlations non-perturbatively within periodic clusters, have also been applied to this model. However, such studies have largely been limited to fixed densities and repulsive interactions. Overall, there have been fewer studies that consider both an extensive range of interaction couplings and band fillings, and relatively less focus on the case of attractive extended interactions.

In this paper, we study the phase diagram of the extended Hubbard model on a square lattice, for both attractive and repulsive nearest-neighbor interactions, using CDMFT+HFD, an extension of the Cluster Dynamical Mean Field Theory (CDMFT) [100, 102] approach with a Hartree-Fock decoupling of the inter-cluster interactions. CDMFT belongs to a class of methods called Quantum Cluster Methods [103–109]. This is a set of approaches that consider a finite cluster of sites embedded in an infinite lattice, and introduce additional fields or "bath" degrees of freedom, determined by variational or self-consistency principles, to best represent the effect of the surrounding infinite lattice. These methods have proven useful for interpolation between results obtained in the weak- and strong-coupling regimes, since their accuracy is controlled by the size of the clusters used, rather than the strength of the couplings. Further, we treat the inter-cluster interactions within a Hartree-Fock mean-field decoupling, which generates additional Hartree, Fock and anomalous contributions to the cluster Hamiltonian. While a similar treatment has been used to study the model at quarter-filling [48] for the case of repulsive interactions, with the objective of understanding the electronic properties of metals close to a Coulomb-driven charge ordered insulator transition, this analysis was focused on a specific parameter regime, and did not include superconducting orders.

This work constitutes a test of the CDMFT+HFD method, described in Sect. 2 below. Our main findings are as follows. For a weak repulsive local interaction $U$ and an attractive extended interaction $V$, the system undergoes a transition towards a phase separated (PS) state when the chemical potential lies in the vicinity of its particle-hole symmetric value, $U/2 + 4V$. The exact region of phase separation is identified by using the hysteresis in the behavior of the electron density as a function of the chemical potential, which corresponds to the coexistence of two different uniform-density solutions. As a function of doping away from the half-filled point, symmetrical and sharply decaying regions of $d_{x^2-y^2}$-wave superconducting order are observed, followed by disconnected regions of extended $s$-wave order near quarter-filling, as well as at very small (large) densities. A stronger attractive extended interaction tends to favor phase separation as well as superconductivity, whereas the repulsive on-site interaction $U$ is found to be detrimental to both. At the particle-hole symmetric chemical potential, we detect a first-order phase transition from antiferromagnetism (AF) to $d$-wave superconductivity as the attractive $V$ becomes stronger, which is accompanied by a gradual deviation of the density from its half-filled limit, induced by phase separation. For repulsive nearest-neighbor interactions in the strong-coupling regime $U \gg t$, the Mott insulating state at half-filling is destabilized, upon hole doping, in favor of a dome-shaped region of $d$-wave superconducting order. This order is found to be remarkably stable in the presence of a non-local interaction, and slightly suppressed by it. At half-filling, a repulsive non-local interaction induces a first-order phase transition from antiferromagnetism (AF) to a charge-density wave (CDW) order. Our results are qualitatively in agreement with the existing literature on the phase diagram of the EHM, with some notable differences in the region of attractive interactions. An important difference is that intra-cluster fluctuations are treated exactly, which tends to make superconducting orders somewhat weaker in this approach.

The paper is organized as follows. In Sect. 2, we introduce the model Hamiltonian, and provide a brief overview of the CDMFT approach that we use for our analysis, as well as the Hartree-Fock mean-field decoupling of the inter-cluster interactions. In Sect. 3, we describe the phase diagram obtained as a function of the interaction strength and doping, and the phase

transitions observed at half-filling. Finally, in Sect. 4, we summarize our results, discuss some relevant observations and present the conclusions of our study.

## 2 Model and method

### 2.1 Model Hamiltonian

The general form of the extended Hubbard model Hamiltonian is

$$H = \sum_{\mathbf{r},\mathbf{r}',\sigma} t_{\mathbf{r}\mathbf{r}'} c^{\dagger}_{\mathbf{r}\sigma} c_{\mathbf{r}'\sigma} + U \sum_{\mathbf{r}} n_{\mathbf{r}\uparrow} n_{\mathbf{r}\downarrow} + \frac{1}{2} \sum_{\mathbf{r},\mathbf{r}',\sigma,\sigma'} V_{\mathbf{r}\mathbf{r}'} n_{\mathbf{r}\sigma} n_{\mathbf{r}'\sigma'}, \tag{1}$$

where $\mathbf{r}, \mathbf{r}'$ label lattice sites, $t_{\mathbf{r}\mathbf{r}'}$ are the hopping amplitudes, $U$ the on-site Hubbard interaction, and $V_{\mathbf{r}\mathbf{r}'}$ the nearest-neighbor interaction (each bond counted once, hence the factor $\frac{1}{2}$).

For the purpose of our analysis, we study the following model on a square lattice:

$$\begin{aligned} H = -t \sum_{\mathbf{r}} \left( c^{\dagger}_{\mathbf{r}} c_{\mathbf{r}+\mathbf{x}} + c^{\dagger}_{\mathbf{r}} c_{\mathbf{r}+\mathbf{y}} + \text{H.c.} \right) + U \sum_{\mathbf{r}} n_{\mathbf{r}\uparrow} n_{\mathbf{r}\downarrow} - \mu \sum_{\mathbf{r}} (n_{\mathbf{r}\uparrow} + n_{\mathbf{r}\downarrow}) \\ + V \sum_{\mathbf{r},\sigma,\sigma'} \left( n_{\mathbf{r}\sigma} n_{\mathbf{r}+\mathbf{x},\sigma'} + n_{\mathbf{r}\sigma} n_{\mathbf{r}+\mathbf{y},\sigma'} \right), \end{aligned} \tag{2}$$

where $\mathbf{x}, \mathbf{y}$ are the lattice unit vectors along the $x$ and $y$ directions, and the operator $c_{\mathbf{r}\alpha}$ annihilates a particle with spin $\alpha = \uparrow, \downarrow$ at site $\mathbf{r}$. The occupation number is $n_{\mathbf{r}\alpha} = c^{\dagger}_{\mathbf{r}\alpha} c_{\mathbf{r}\alpha}$. We consider a range of values for the chemical potential $\mu$, corresponding to a continuous range of densities, from $n = 0$ to 2, along with a repulsive local interaction $U > 0$, and a nearest-neighbor interaction $V$ that can be positive or negative. The particle-hole symmetric value of the chemical potential, $\mu = U/2 + 4V$, which corresponds to a half-filled band in the absence of phase separation, features prominently in our analysis. The unit of energy is taken to be the nearest-neighbor hopping amplitude $t = 1.0$, with the lattice constant $a = 1$. Note that in the absence of longer-range hopping terms, beyond the nearest-neighbor bonds, the model respects particle-hole symmetry $n \to 2 - n$.

We examine the possibility of superconducting as well as density-wave orders. For this purpose, the anomalous operators are defined on the lattice using a $d$-vector, as

$$\Delta_{\mathbf{r}\mathbf{r}',b} c_{\mathbf{r}s} (i\sigma_b \sigma_2)_{ss'} c_{\mathbf{r}'s'} + \text{H.c.}, \tag{3}$$

where $b = 0, 1, 2, 3$, and $\sigma_b$ are the Pauli matrices. The case $b = 0$ corresponds to singlet superconductivity, in which case $\Delta_{\mathbf{r}\mathbf{r}',0} = \Delta_{\mathbf{r}'\mathbf{r},0}$ and the cases $b = 1, 2, 3$ correspond to triplet superconductivity, in which case, $\Delta_{\mathbf{r}\mathbf{r}',b} = -\Delta_{\mathbf{r}'\mathbf{r},b}$. In practice, these operators are defined by specifying $b$ and the relative position $\mathbf{r} - \mathbf{r}'$.

Density wave operators are defined with a spatial modulation characterized by a wave vector $\mathbf{Q}$, and can be based on sites or on bonds. In our analysis, we focus on site density waves, defined as

$$\sum_{\mathbf{r}} A_{\mathbf{r}} \cos(\mathbf{Q} \cdot \mathbf{r} + \phi), \tag{4}$$

where $A_{\mathbf{r}} = n_{\mathbf{r}}, S^x_{\mathbf{r}}, S^z_{\mathbf{r}}$ corresponds to charge- or spin-density wave orders, and $\phi$ is a sliding phase. We probe the presence of density-wave orders with $\mathbf{Q} = (\pi, \pi)$ and $\phi = 0$.

### 2.2 Method: CDMFT+HFD

Let us briefly describe the method used in our analysis, Cluster dynamical mean-field theory (CDMFT). For a detailed discussion of the basic principles of such Quantum Cluster Methods, please see Ref. [103, 105, 110].

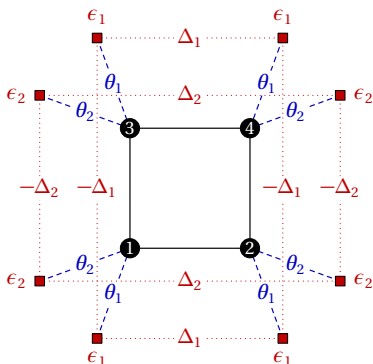

Figure 1: Schematic representation of the first ("simple") impurity problem used in our analysis, with bath energies $\epsilon_i$, cluster-bath hybridization parameters $\theta_i$ and anomalous bath parameters $\Delta_i$. Physical sites are marked by numbered black dots and bath orbitals by red squares. We choose the bath parameters such that the environment of each cluster site is identical. This impurity model has reflection symmetry with respect to horizontal and vertical mirror planes ($C_{2v}$ symmetry), and typically involves only spin-independent hopping terms. Pairing terms $\Delta_{1,2}$ are introduced between bath orbitals, with signs adapted to the SC order probed (shown here for a $d$-wave order, but all positive for an extended $s$-wave order). The number of independent bath parameters is 6.

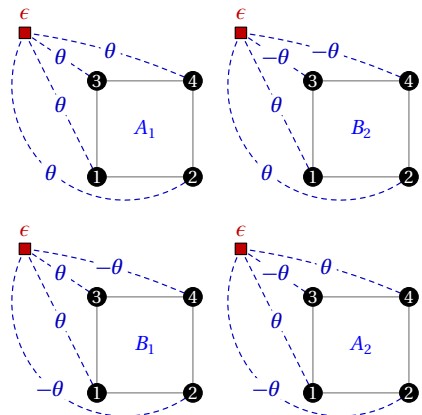

Figure 2: Schematic representation of the second ("general") impurity problem used in our analysis. Each representation of the point group $C_{2v}$ ($A_{1,2}$ and $B_{1,2}$) corresponds to a set of phases ($\pm 1$), and each of the 8 bath orbitals belongs to one of these four representations (two bath orbitals per representation). The different bath orbitals are independent (the bath system is diagonal) and we only show here a view of each of the four representations with the corresponding signs associated to each cluster site (black dots). The hybridization parameters $\theta$ are shown, and corresponding pairing operators (or anomalous hybridizations) between each bath orbital and each site also exist, with the same relative phases. We have 3 parameters per bath orbital, which leads to a total of 24 bath parameters, and subtracting six constraints due to a $C_{4v}$ rotational symmetry, we obtain 18 independent bath parameters for the general model.

This approach is an extension of dynamical mean-field theory (DMFT) [111–114], which accounts for short-range spatial correlations, by considering a cluster of sites with open boundary conditions, instead of a single-site impurity. The effect of the cluster's environment is taken into account by introducing a set of uncorrelated "bath" orbitals hybridized with it. In this manner, the infinite lattice is tiled into identical clusters coupled to a bath of auxiliary, uncorrelated orbitals, with energy levels $\epsilon_{i\sigma}$, which may or may not be spin dependent, and hybridized with the cluster sites (labeled $r$) with amplitudes $\theta_{ir\sigma}$. In addition, for studying superconducting orders, different types of anomalous pairings $\Delta_{ij\sigma\sigma'}$ may be introduced between bath orbitals $i, j$ or $\Delta_{ir\sigma\sigma'}$ between bath orbital $i$ and cluster site $r$.

The cluster and bath size is limited by the exact diagonalization solver: the practical upper limit for the total number of cluster and bath orbitals is 4+8=12, given that the ground state and Green function must be computed repeatedly in this approach. A true finite-size analysis is impossible here, for the next cluster size of the same square geometry would be 9, and the number of bath orbitals would need to grow accordingly. Even in a one-dimensional model, analyzing finite-size effects in CDMFT is challenging, because of the combined effects of cluster size and bath size [115].

We use two types of bath models. In the simple model (Fig. 1), the environment of each cluster is identical, and we introduce two bath orbitals per cluster site. Parameters of the impurity model include bath orbital energy levels ($\epsilon_{1,2}$), hybridization between each cluster site and the corresponding bath orbitals ($\theta_{1,2}$), and pairings between the bath orbitals ($\Delta_{1,2}$). The precise form of $\Delta_{1,2}$, including their relative phases between different bath orbitals, depends on whether we probe extended $s$-wave, $d$-wave, or triplet superconductivity. This simple impurity model involves 6 independent parameters to be determined self-consistently. At half-filling, we introduce bath energies as well as hoppings, that are consistent with the appearance of a density-wave order, and additionally spin-dependent in the presence of antiferromagnetism. This increases the number of independent parameters. However, imposing particle-hole symmetry at half-filling once again reduces this number to 6. For $V < 0$, we do not impose particle-hole symmetry on the bath parameters due to the possibility of phase separation, and the number then increases to 10.

We also use a more general bath model (Fig. 2). While the total number of bath orbitals is unchanged, every bath orbital is connected to every cluster site (with distinct combinations of relative phases), and we define bath energies, cluster-bath hybridizations and anomalous pairings between the cluster and the bath sites. In this model the bath is diagonal, i.e., the different bath orbitals are not directly coupled between themselves. We have 3 parameters per bath orbital, and taking into account six constraints due to rotation symmetry, there are 18 independent bath parameters to set. At the particle-hole symmetric chemical potential, we introduce bath energies, hybridizations and anomalous pairings that have two different values for alternative sites. This gives us a total of 42 independent parameters in the absence of particle-hole symmetry for $V < 0$ and 15 independent parameters when superconductivity is absent (i.e. for $V > 0$) and particle-hole symmetry is taken into account.

All bath parameters are determined by a self-consistency condition (see Ref. [103,105,110] for details). The simple bath model is expected to be easier to converge than the general bath model, because of the smaller set of parameters. While we expect the results obtained from the general bath model to be more reliable, we do find most of the results to be qualitatively similar in the two cases. Once the bath parameters are converged, the self-energy $\Sigma(\omega)$ associated with the cluster is applied to the whole lattice, so that the lattice Green function is

$$\mathbf{G}^{-1}(\tilde{\mathbf{k}}, \omega) = \mathbf{G}_0^{-1}(\tilde{\mathbf{k}}, \omega) - \Sigma(\omega). \tag{5}$$

Here, $\tilde{\mathbf{k}}$ denotes a reduced wave vector (defined in the Brillouin zone of the super-lattice of clusters defined by the tiling) and $\mathbf{G}_0$ is the non-interacting Green function. The Green-

function-like objects $\mathbf{G}$, $\mathbf{G}_0$ and $\Sigma$ are $L \times L$ matrices, $L$ being the number of physical degrees of freedom on the cluster (here $L = 8$ because of spin and the four cluster sites). The average values of one-body operators defined on the lattice are obtained using the lattice Green function $\mathbf{G}$ determined from the solution for the optimum values of the bath parameters. An exact-diagonalization solver (the Lanczos method or variants thereof) is used at zero temperature. The computational size of the problem increases exponentially with the total number of cluster and bath orbitals.

In the presence of extended interactions, we also perform a Hartree-Fock mean-field decomposition of the interaction terms defined between different clusters, while the interactions within a cluster are treated exactly. The inter-cluster interactions are decoupled in the Hartree, Fock and anomalous channels, which contribute to the number density, the hopping and the pairing operators, respectively. Moreover, we only retain those combinations of the site/bond operators that are physically relevant in the regions we work in (such as $d$-wave or extended $s$-wave), and discard the rest. The mean-field values of the relevant combinations are determined self-consistently, within the CDMFT loop that optimizes the bath parameters. For the details of this procedure, please refer to Appendix A. For a comparison of different methods used for solving the self-consistent nonlinear equations involved in the CDMFT procedure, please refer to Appendix B.

# 3 Results

In this section, we discuss the salient features of the phase diagram obtained from our analysis, for both attractive and repulsive nearest-neighbor interactions. The dominant superconducting and density-wave orders are identified by computing the corresponding order parameters using the optimum values of the CDMFT (bath and mean-field) parameters, as a function of electron density, as well as at half-filling. In the following analysis, we focus our attention on the strong coupling limit $U \gg t$ for $V > 0$, which is a regime well-understood on physical grounds. For $V < 0$, we consider relatively weak interactions $U \sim t$, far from the Mott insulating regime, which primarily serve the purpose of controlling the extent of phase separation when the interaction $V$ becomes sufficiently attractive. At half-filling, we confirm the nature of the phase transitions, by plotting the relevant order parameters both as a function of $U > 0$, for fixed values of $V > 0$ or $V < 0$, and as a function of $V$ for fixed values of $U$.

## 3.1 Phase diagram at the particle-hole symmetric chemical potential

Here, we fix the chemical potential to $\mu = U/2 + 4V$, corresponding to a half-filled band, and examine the behavior of different superconducting and density-wave orders, as a function of the local repulsion $U$ as well as attractive/repulsive $V$. While antiferromagnetism is favored at half-filling, in both weak- and strong-coupling regimes, an attractive non-local interaction is expected to drive the system towards a superconducting instability, and eventually phase separation. On the other hand, repulsive interactions $V$ would typically foster competition between charge and spin fluctuations, and favor a charge-ordered state. Below, we discuss the results obtained using the simple bath model (Fig. 1).

### 3.1.1 $V < 0$

For a fixed attractive nearest-neighbor interaction $V$, as the strength of the local repulsive interaction $U$ decreases, the system undergoes a first-order phase transition from antiferromagnetism to $d$-wave superconductivity. This is accompanied by a deviation in the electron

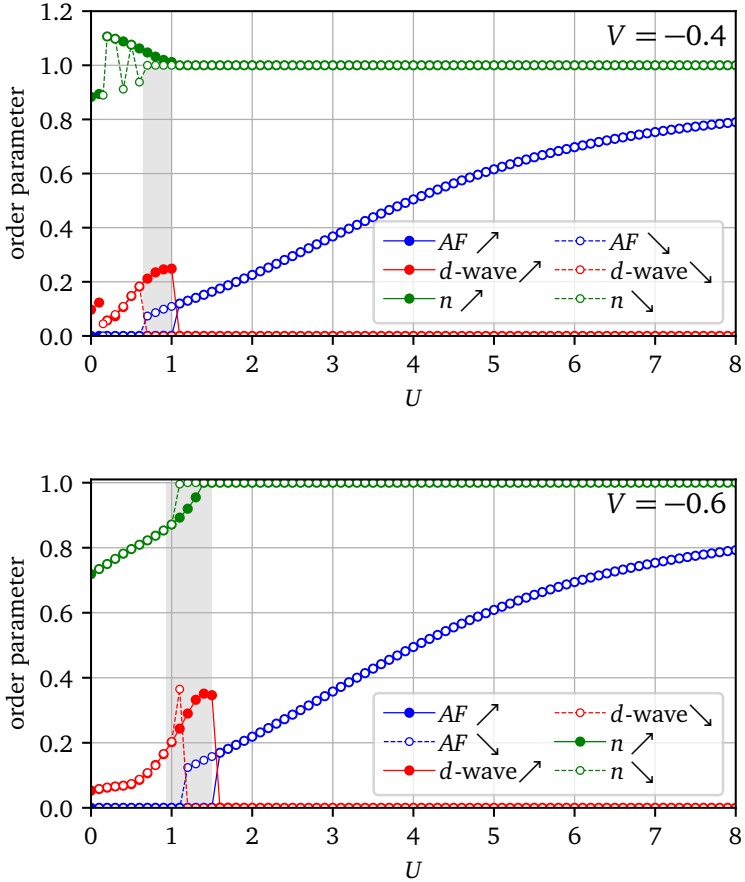

Figure 3: First-order phase transition from $d$-wave superconductivity (indicated by filled/open red circles) to antiferromagnetism (AF, indicated by filled/open blue circles), as a function of the repulsive local interaction $U$, at fixed $V = -0.4$ (top) and $V = -0.6$ (bottom), and fixed chemical potential $\mu = U/2 + 4V$ (particle-hole symmetric point). The simple impurity model (Fig. 1) is used. The transition is accompanied by a deviation in the number density (indicated by filled/open green circles) from the half-filled value $n = 1$, meaning that we are entering a phase separated regime. The dashed (solid) curves of each color depict the behavior of the different quantities for decreasing (increasing) $U$, respectively. The prominent region of hysteresis between the two curves confirms the order of the transition. The small jump/discontinuity observed in the $d-$wave order parameter for increasing $U$ for $V = -0.4$ results from issues with the convergence of the CDMFT procedure at that point. On the other hand, for $V = -0.6$, we observe a jump in the $d-$wave order parameter for decreasing $U$, which appears to signal a transition from a $d-$wave order at half-filling to one coexisting with phase separation, rather than being a numerical error. Likewise, for increasing $U$, we observe a nontrivial $d-$wave order parameter both in the presence and absence of phase separation for $V = -0.6$ (for more details, see Appendix B).

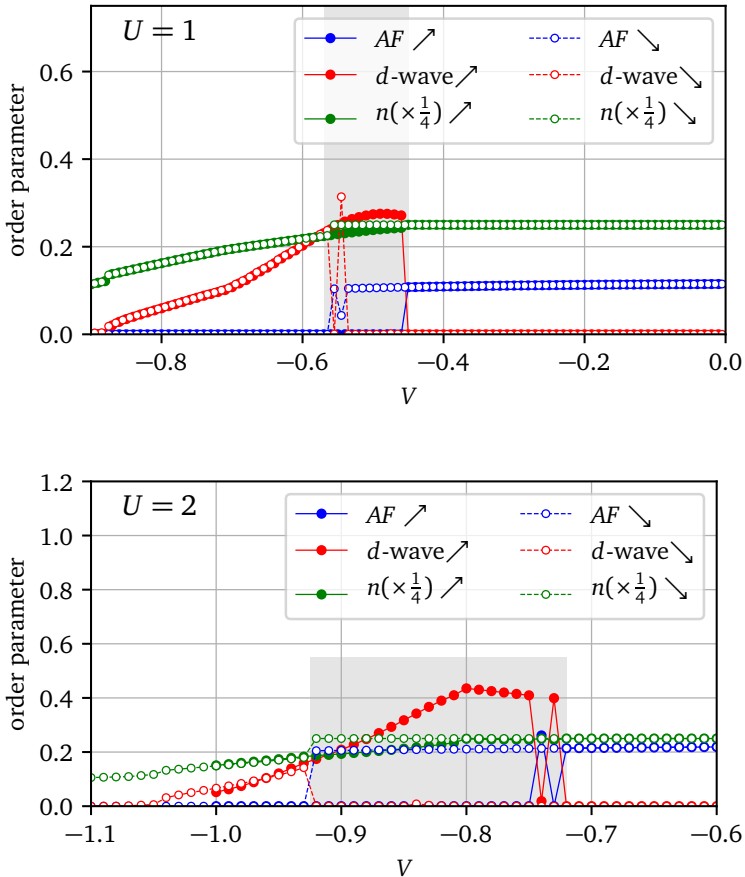

Figure 4: First-order phase transition from antiferromagnetism (AF) (indicated by filled/open blue circles) to $d$-wave superconductivity (indicated by filled/open red circles), for increasingly attractive $V$, followed by a rapid suppression in the superconducting order parameter, for on-site interaction $U = 1$ (top) and $U = 2$ (bottom). The simple impurity model (Fig. 1) is used. The transition is accompanied by a deviation in the number density (indicated by filled/open green circles) from the half-filled value $n = 1$. The dashed (solid) curves of each color depict the behavior of different quantities for decreasing/more negative (increasing/less negative) $V$, and we find significant hysteresis. For larger repulsive interactions $U$, the transition is found to occur at a critical value of $V$ that is more attractive. For $U = 1$, we observe oscillations between the $d-$wave and AF orders at half-filling, close to the transition for decreasing/more negative $V$, while for $U = 2$, we see a significant region of $d-$wave superconductivity close to half-filling for increasing/less negative $V$, as well as similar oscillations between the $d-$wave and AF orders at half-filling, close to the transition between the two states for increasing/less negative $V$.

density from its half-filled limit, which can be attributed to the effects of phase separation, discussed in more detail in the next subsection. Each of the order parameters is plotted for both increasing and decreasing $U$, and the region of hysteresis between the two curves indicates that the transition is first-order in nature. We have verified that other pairing symmetries, such as extended $s$-wave and $p$-wave, do not compete with $d_{x^2-y^2}$ pairing in this regime. The results of our analysis are shown in Fig. 3. Likewise, an antiferromagnetic order is destabilized in favor of $d$-wave superconductivity for an attractive $V$, at a fixed repulsive $U \sim t$, with signif-

icant hysteresis between the curves obtained for increasing/decreasing $V$. The latter state is then rapidly suppressed due to the effect of phase separation. The results are shown in Fig. 4.

### 3.1.2   $V > 0$

For repulsive nearest-neighbor interactions $V$, we do not expect to find any superconducting orders at half-filling in the strong-coupling limit $U \gg t$, and instead focus on studying the competition between charge- and spin-density-wave orders. At fixed $V > 0$, we observe a first-order phase transition from a charge-density wave (CDW) to an antiferromagnetic (AF) state, as a function of increasing $U$. Likewise, for a large repulsive $U$, the system undergoes a phase transition from antiferromagnetism to CDW, as a function of the repulsive $V$. In both cases, a large region of hysteresis is observed between the results obtained for increasing and decreasing values of the respective interaction couplings. The results of our analysis are shown in Figs 5 and 6, respectively.

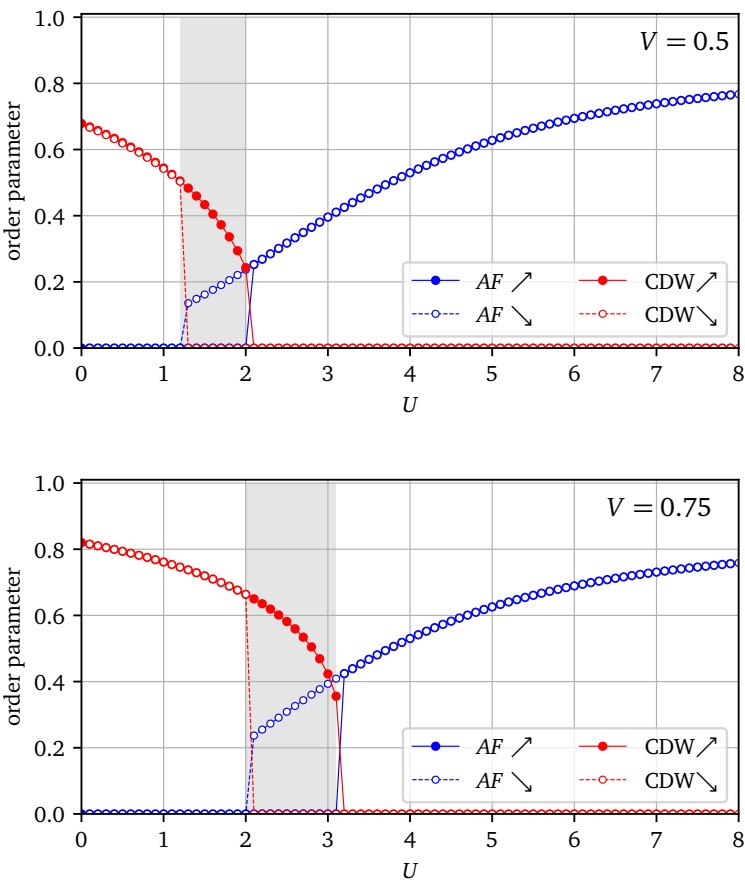

Figure 5: First-order phase transition from a charge-density wave (CDW) order (indicated by filled/open red circles) to antiferromagnetism (indicated by filled/open blue circles), at half-filling, as a function of the local repulsive interaction $U$, for $V = 0.5$ (top) and $V = 0.75$ (bottom). The simple impurity model (Fig. 1) is used. The dashed (solid) curves of each color depict the behavior of the order parameters for decreasing (increasing) $U$, and exhibit significant hysteresis. As the repulsive $V$ becomes stronger, the transition is found to occur at a larger value of $U$, the CDW order parameter increases considerably in magnitude, and the region of hysteresis is somewhat enhanced.

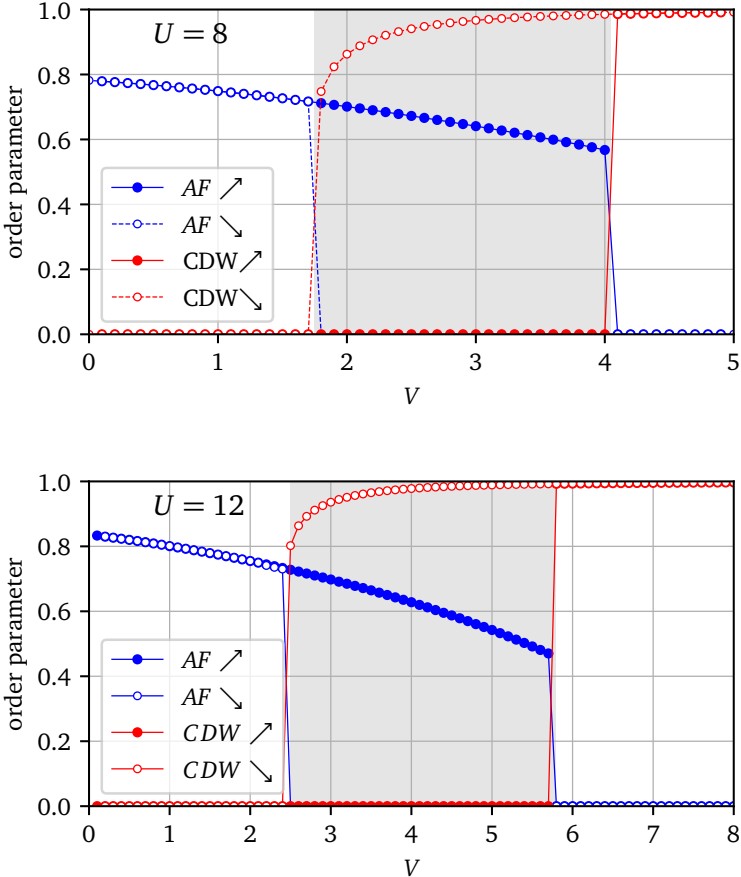

Figure 6: First-order phase transition from antiferromagnetism (indicated by filled/open blue circles) to charge-density wave (CDW) order (indicated by filled/open red circles), at half-filling, as a function of the repulsive interaction $V$ for fixed $U$, with $U = 8$ (top) and $U = 12$ (bottom). The simple impurity model (Fig. 1) is used. The dashed (solid) curves of each color depict the behavior of the order parameters for decreasing (increasing) $V$, and exhibit considerable hysteresis. As $U$ increases, the transition occurs at a larger critical value of $V$, and the antiferromagnetic order parameter increases in magnitude.

We do not present the corresponding results for the more general bath model (Fig. 2) here, as they are found to be qualitatively similar to those obtained for the simple model. The key differences, that are sometimes observed, include a) an increase/decrease in the strength of the $d$-wave order parameter close to the transition, b) a smaller region of hysteresis, c) a small shift in the position of the transition, particularly as a function of $V$ for fixed $U$.

## 3.2 Phase diagram as a function of density

Next, we examine the phase diagram of the model over a continuous range of densities, for $U > 0$ and attractive/repulsive $V$. For $V > 0$, we once again limit ourselves to the strong-coupling limit $U \gg t$. For $V < 0$, we focus on studying the effect of an attractive extended interaction, with a local repulsion $U$ controlling the extent of phase separation.

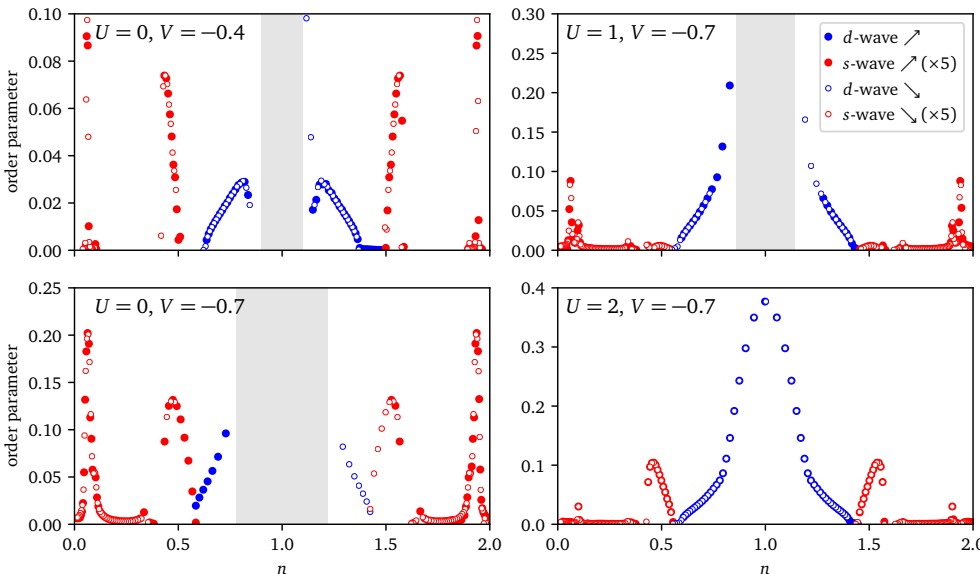

Figure 7: Superconducting order parameter of the EHM with attractive nearest-neighbor interactions, as a function of density $n$, from $n = 0$ to 2 for the simple bath model (Fig. 1). Close to the half-filled value $n = 1$, we find signatures of phase separation, indicated by a gap in the curve over a range of densities, caused by a jump in the compressibility $\partial n / \partial \mu$ (as shown in Fig. 9). For smaller (larger) fillings, nearly symmetrical and sharply defined regions of $d$-wave superconductivity (represented by filled/open blue circles) are followed by disconnected patches of extended $s$-wave order (represented by filled/open red circles), which appear only beyond a critical attractive value of $V$. Note that the asymmetry between either the $d-$wave regions or the extended $s$-wave regions near the band edges, especially evident for $V = -0.4$, is a numerical artefact owing to insufficient accuracy in the CDMFT procedure and has no physical consequence.

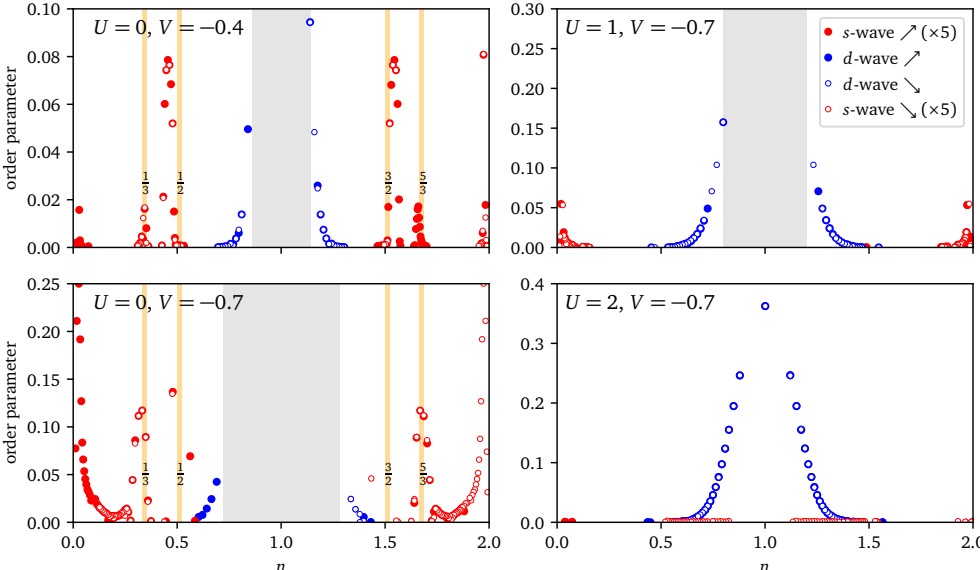

Figure 8: Superconducting order parameter of the EHM with attractive nearest-neighbor interactions, as a function of density $n$, from $n = 0$ to 2 for the general bath model (Fig. 2). The overall behavior of the $d-$ and extended $s$-wave patches are similar to the corresponding result for the simple bath model. However, note that the structure of the $s$-wave order parameter has changed, with a more extended region near quarter-filling, and an additional patch near $1/3-$filling. For $U = 0, V = -0.7$, the phase separation region extends all the way to quarter-filling, and the corresponding superconducting patches are almost absent, and asymmetric about $n = 1$. Moreover, the new $s$-wave order parameter becomes unambiguously weaker as the repulsive $U$ increases, and is completely absent for $U = 1$ and $U = 2$, thus resolving the question of the non-monotonous behavior of the $s$-wave order parameter in the simple bath model.

### 3.2.1 $V < 0$

Let us now discuss the different phases that are supported by the model as a function of density. Close to half-filling, we find a region of phase separation, indicated by a jump in the density, flanked by symmetrical islands of $d_{x^2-y^2}$ pairing, which decay rapidly as a function of density. For further smaller (larger) fillings, an extended $s$-wave order appears in the form of disconnected regions, near quarter-filling and at very small (large) densities. Interestingly, the variation of the extended $s$-wave order parameter as a function of $U$ and $V$ are found to be different for the simple bath model and the more general one. In the case of the simple model (see Fig. 7), we find small regions of extended $s$-wave superconductivity near quarter-filling, that vary non-monotonously as a function of $U$. Only for sufficiently attractive $V$, nearly symmetrical regions of extended $s$-wave order also appear close to the band edges. The corresponding results for the general bath model are illustrated in Fig. 8. While the overall magnitude of the $s$-wave order parameter turns out to be smaller than in the previous case, its shape is more extended at quarter-filling, with two patches appearing next to each other, which, interestingly, appear close to fillings of $1/3$ and $1/2$, respectively. While it is tempting to blame the $n = 1/2$ feature on a commensurate finite-size effect on a 4-site cluster, this is less obvious for the $n = 1/3$ feature. The superconductivity also clearly becomes stronger as a function of $V < 0$. Notably, the $s$-wave order is clearly absent for both $U = 1$ and $U = 2$, thus eliminating the confusion caused by the aforementioned non-monotonous variation in the case of the sim-

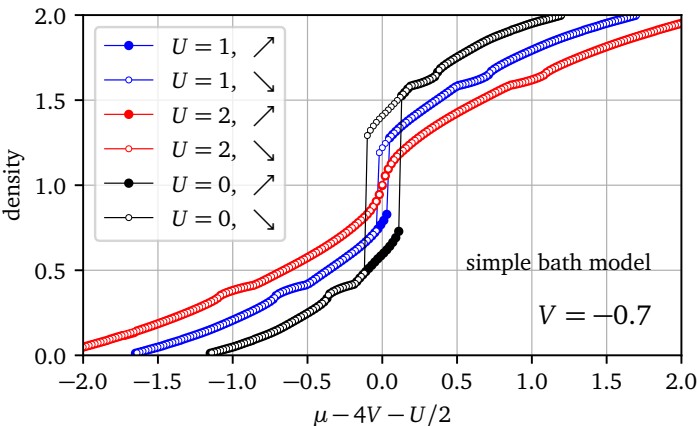

Figure 9: Number density $n$ as a function of the chemical potential $\mu$ (measured with respect to its particle-hole invariant value, $\mu_c = U/2 + 4V$) for an EHM with attractive nearest-neighbor interactions, over a range of values of $U \geq 0$ and $V < 0$ for the simple bath model (Fig. 1). On either side of half-filling ($\mu = \mu_c$), we find symmetrical jumps in the compressibility $\partial n / \partial \mu$ enclosing a region of hysteresis, which corresponds to the coexistence of two different uniform-density solutions. This is interpreted as the region of phase separation. The red, blue and black filled/open circles represent the behavior for various values of $U$ for $V = -0.7$, and demonstrate that while a sufficiently attractive interaction $V$ favors phase separation, a stronger on-site repulsion $U$ is detrimental to it.

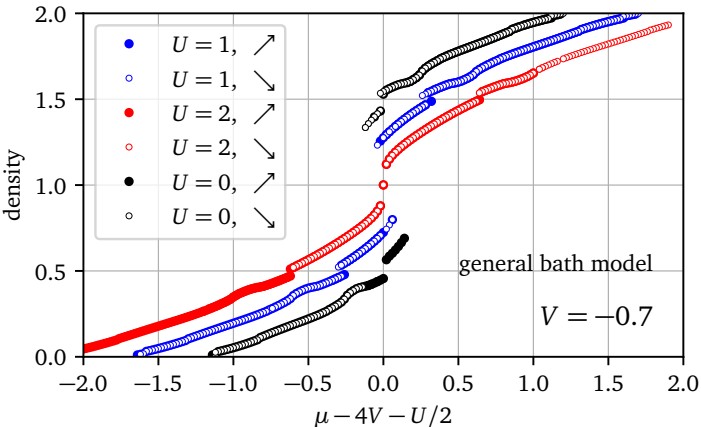

Figure 10: Number density $n$ as a function of the chemical potential $\mu$ (measured with respect to its particle-hole invariant value, $\mu_c = U/2 + 4V$) for the EHM with attractive nearest-neighbor interactions, over a range of values of $U \geq 0$ and $V < 0$ for the general bath model (Fig. 2). The behavior is very similar to that observed in the simple bath model, with the most notable difference being the appearance of symmetric jumps in the number density $n$, close to quarter-filling, for each of the curves.

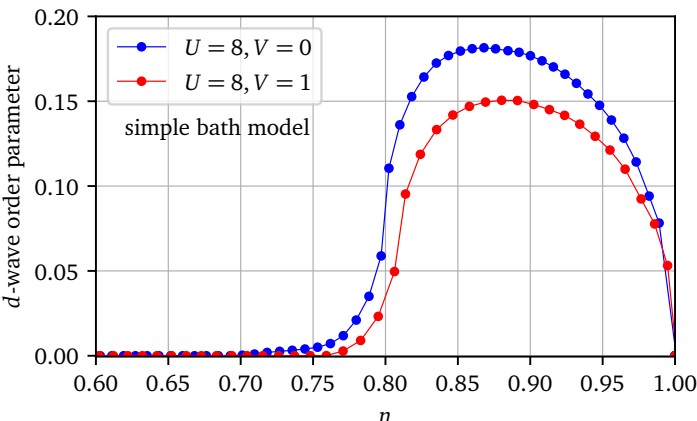

Figure 11: Superconducting $d$-wave order parameter of the EHM with repulsive nearest-neighbor interactions in the strong-coupling limit, i.e., at $U = 8t$, using the simple bath model (Fig. 1). The Mott insulating state at half-filling is destabilized in favor of $d_{x^2-y^2}$ pairing, upon hole doping. The dome-like region of $d$-wave superconducting order is observed for $V = 0$ (indicated by the solid blue curve) and is somewhat suppressed for nonzero repulsive $V$ (indicated by the solid red curve). No other superconducting orders are found to be stabilized in this region.

ple model, and illustrating the advantage of considering a larger number of bath parameters in the CDMFT procedure. This being said, the conclusions from the two bath models are very similar. Using two different bath models provides us with an order-of-magnitude estimate of the error caused by the discreteness of the bath.

To better characterize the region of phase separation, we examine the behavior of the number density $n$ as a function of the chemical potential $\mu$, measured with respect to its particle-hole symmetric value $\mu_c = U/2 + 4V$. On either side of $\mu = \mu_c$, we find symmetrical jumps in the compressibility $\partial n / \partial \mu$, enclosing a region of hysteresis in the $\mu - n$ curve, depicted in Fig. 9, where two uniform-density solutions coexist. Within our approach, this is interpreted as the region of phase separation, and is found to shrink under the influence of stronger local repulsive interactions $U$, and expand when $V$ becomes more attractive. The corresponding results for the general bath model are depicted in Fig. 10. The two sets of results are qualitatively similar, except for symmetric jumps observed in the number density $n$ near quarter-filling in the latter case. We note that the jumps occur only for the model with the larger number of bath parameters, and are the most prominent for $U = 0, V = -0.7$, where the phase separation region extends all the way to quarter-filling, becoming progressively smaller for $U = 1$ and 2. It is plausible that phase separation might lead to the appearance of multiple jumps in the density, at half-filling as well as quarter-filling. Moreover, a finite-size effect would have been even more obvious in the simple bath model, where these jumps are found to be absent. The origin of the jumps is currently unclear to us.

The appearance of a phase separated state for sufficiently attractive interactions is a familiar result [32, 52, 71, 81, 87, 116], which has received attention from other groups, including very recently [70], but the characterization of the region of phase separation tends to depend on the method used for the analysis, and whether it is capable of handling a nonuniform distribution of particles.

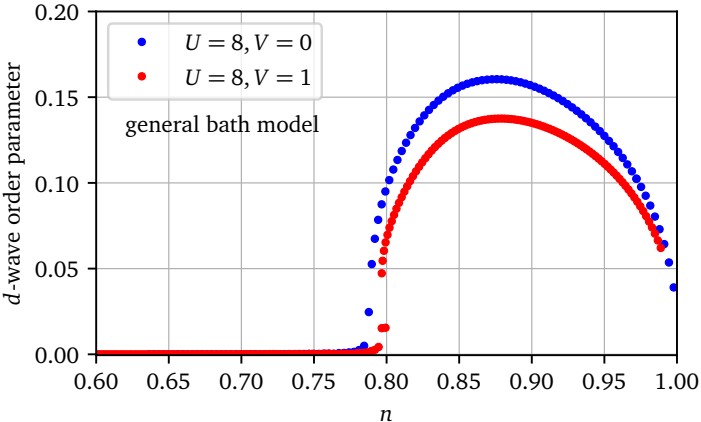

Figure 12: Superconducting $d$-wave order parameter of the EHM with repulsive nearest-neighbor interactions in the strong-coupling limit ($U = 8t$) using the general bath model (Fig. 2). The behavior is qualitatively similar to that obtained in the simple model, with a slight difference in the magnitudes of the $d$-wave order parameter. The most noticeable difference between the two bath models is the relatively sharp transition into and out of the $d$-wave superconducting phase.

### 3.2.2 $V > 0$

At half-filling, for $U = 8t$, the large on-site interaction freezes the charge degree of freedom, and the ground state is a Mott insulator. Hole doping is found to destabilize the magnetic order, and drive the system towards a $d$-wave superconducting phase. We encounter a dome-shaped region of $d$-wave superconductivity for $V = 0$, which is suppressed at smaller densities, where no competing superconducting orders are found to be stabilized in our analysis. Upon introducing a repulsive $V \sim t$, the superconducting order remains stable, but is somewhat suppressed. The results are depicted in Fig. 11. The corresponding results for the general bath model are depicted in Fig. 12. The two sets of results are qualitatively similar, with the most noticeable difference being the relatively sharper transition to and from the $d$-wave ordered state in the latter case. These results are consistent with the picture of superconductivity mediated by short-range spin fluctuations in a doped Mott insulator [117–119].

## 4 Discussion and conclusions

In summary, we have studied the phase diagram of the extended Hubbard model, for both attractive and repulsive nearest-neighbor interactions, using a combination of Cluster Dynamical Mean Field Theory (CDMFT), with a dynamical Hartree-Fock approximation for treating inter-cluster interactions. We examine possible phase transitions at half-filling, as well as the dominant phases that are stabilized as a function of density. At the particle-hole invariant chemical potential, which corresponds to a half-filled band in the absence of phase separation, the antiferromagnetically ordered state undergoes a first-order phase transition to $d$-wave superconductivity for a critical attractive interaction $V$. Stronger attractive extended interactions also tend to induce phase separation, which manifests itself in the form of a gradual deviation of the density from its half-filled limit, for a fixed chemical potential. For a sufficiently strong repulsive interaction $V$, a charge-density wave order is stabilized at half-filling.

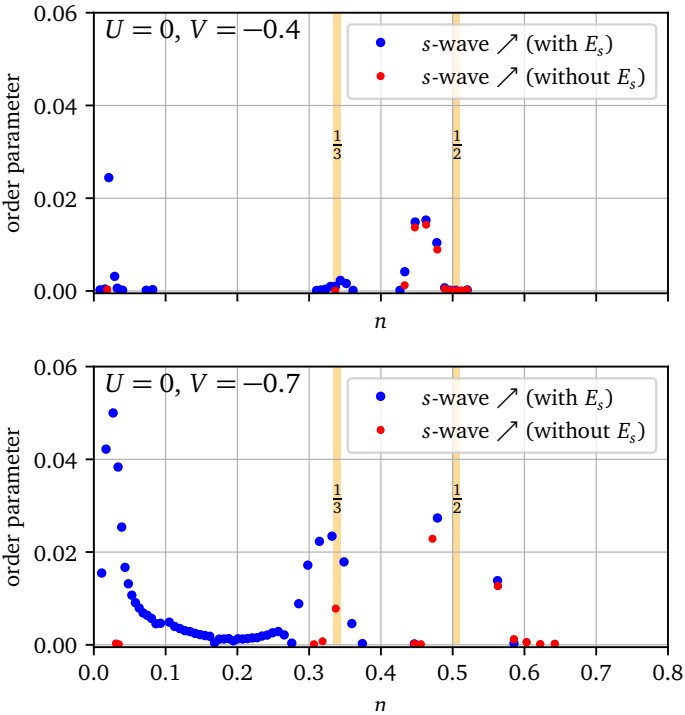

Figure 13: The figure shows the behavior of the extended $s$-wave order parameter as a function of the number density $n$, with and without the inclusion of the self-consistent anomalous mean-field parameter $E_s$ (see Appendix A), for $U = 0, V = -0.4$ (above) and $U = 0, V = -0.7$ (below). Clearly, some of the regions with a nontrivial $s$-wave order parameter are found to be absent when $E_s$ is not included. For $U = 0, V = -0.7$, the most prominent among these appears to be the region with density in the range $0 < n < 0.3$. Upon considering a stronger attractive $V$, these regions tend to reappear, but are suppressed in magnitude in the absence of $E_s$.

As a function of density, a phase separated state near the half-filled point is flanked by symmetrical regions of $d$-wave superconductivity, that decay sharply as a function of density, and islands of extended $s$-wave order at smaller (larger) band fillings. For the case of repulsive non-local interactions, in the strongly coupled limit, the Mott insulator at half-filling gives way to a dome-shaped region of $d$-wave superconductivity, upon hole doping, which is expected on physical grounds. No other competing superconducting orders are found to be stabilized in this region of parameter space.

For the most part, our results are found to be qualitatively consistent with the existing literature. The transition between antiferromagnetism and CDW at half-filling, for repulsive interactions, has been predicted by several previous studies [26,31,54,58,62,65,70,76–78,87], although the critical interaction strength typically depends on the method of analysis. For densities away from half-filling, there have also been some predictions of $d_{xy}$ pairing, that appears beyond the region of $d_{x^2-y^2}$ pairing, for repulsive extended interactions [39,56]. We do not find such a state in our analysis. The phase diagrams obtained from self-consistent mean-field theory based analyses tend to prominently feature $d$-wave superconductivity at half-filling, with a continuous region of extended $s$-wave order at smaller densities, along with a region of coexistence between the two, i.e., $s + id$ pairing [50,51]. In our analysis, we do not usually see a coexistence between $d$- and extended $s$-wave orders. In the simple model, such a coexistence is observed only in those regimes where both interactions $U > 0$ and $V < 0$ are sufficiently strong, and comparable in magnitude. This may be due to the

fact that the superconducting orders found in our analysis are fairly weak, and the significant attractive interactions that are, therefore, needed for stabilizing overlapping regions of $d$- and extended $s$-wave orders, would also lead to a larger region of phase separation. This effect can only be compensated by including a sufficiently large repulsive local interaction. On the other hand, we have not been able to verify a similar coexistence of the orders for the general bath model, due to the rapid suppression of the extended $s$-wave order, near quarter-filling, with an increase in $U$. Some studies have also suggested the possibility of $p$-wave superconductivity, especially at half-filling [32], and for intermediate hole doping, beyond the region of $d$-wave superconducting order [39, 50, 51]. We do not find signatures of $p$-wave superconductivity in the parameter regimes that we study. Some of our results at half-filling are found to be qualitatively consistent with a recent study on the extended Hubbard model using the determinantal Quantum Monte Carlo technique [70], which also reports the transitions between $d$-wave superconductivity and AFM, as well as between phase separation and $d$-wave, that we observe in our analysis. In addition, the authors of the aforementioned paper also explore other quadrants of the $U - V$ phase diagram, including the case where $U < 0$, which we do not take into account, since the repulsive component of the Coulomb interaction is always expected to be present in a realistic situation.

In contrast to ordinary mean-field theory, our approach takes the intra-cluster fluctuations into account exactly, and is therefore expected to give more reliable quantitative results. In particular, ordered phases are weaker in this approach than in ordinary mean-field theory. At the same time, it should be noted that we only take into account spatial fluctuations within small clusters, and the accuracy of the method is controlled by the size of the clusters used. To illustrate the importance of including the effect of the inter-cluster interactions self-consistently, which are usually disregarded in cluster-based approaches, we have compared the behavior of the superconducting $d$- and extended $s$-wave orders as a function of density $n$, for an attractive $V$ (see Fig. 13) in the presence and absence of the anomalous mean-field parameters (which we refer to as $E_d$ and $E_s$ respectively). Certain regions of the extended $s$-wave order, that we observe in our analysis, disappear entirely in the absence of the self-consistent anomalous mean field parameter $E_s$. These regions tend to reappear, but with a smaller amplitude, when the attractive $V$ is sufficiently strong. Likewise, in the case of $d$-wave superconductivity, we find that the superconducting order parameter is negligible when $E_d$ is absent, and tends to reappear, with a much smaller amplitude, when the repulsive $U$ is increased. Our approach is more suitable for making predictions about the thermodynamic limit than exact diagonalization studies on finite-sized clusters, since only the self-energy is limited by the cluster size. Some recent studies have explored the possibility of magnetic states characterized by ordering wave vectors that are incommensurate with the lattice periodicity [120] in the two-dimensional Hubbard model, for electron densities below half-filling, where the antiferromagnetic state becomes unstable. Our approach is unsuitable for identifying such incommensurate charge and spin orders. Our method does not suffer from fundamental restrictions on its applicability in any particular parameter regime, and allows us to study the behavior of the model as a continuous function of doping, rather than by focusing on specific densities, as has been done in many previous studies. In the future, this method could be potentially useful for analyzing more complicated models, including those with spin-orbit interactions. It can also be applied to the single-band Hubbard model on a triangular lattice, in which the importance of non-local interactions has been pointed out in the literature [121]. It would also be interesting to explore the regime of non-perturbative repulsive local interactions and attractive extended interactions, to observe their combined effect on driving or suppressing phase separation [122, 123]. Longer-range hopping terms can also be included within our exact diagonalization implementation, which give rise to geometric frustration, making the analysis more relevant for the physics of the cuprates.

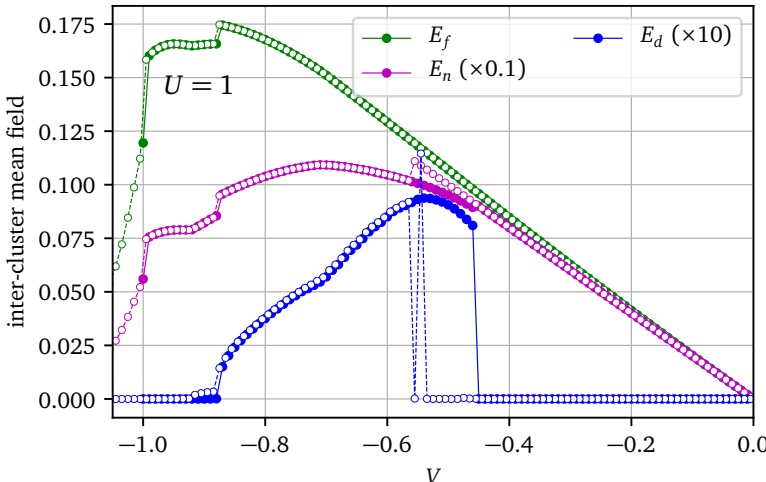

Figure 14: Inter-cluster Hartree-Fock mean fields for the solutions shown in the top panel of Fig. 4. $E_d$ is the eigen-operator associated with $d$-wave superconductivity, $E_f$ with the nearest-neighbor kinetic operator $f_{\mathbf{rr}'\sigma\sigma}$ and $E_n$ with the density $n$ (basically a shift in the chemical potential induced by $V$). The mean-field $E_s$ associated with extended $s$-wave superconductivity is negligible over almost the entire range of $V$, since this is at half-filling, except at significantly attractive $V$ (due to phase separation). Note the very different scales (the superconducting mean field is much magnified). The filled and empty circles denote the results for increasing (less negative) and decreasing (more negative) $V$, respectively. The oscillations in the $d-$wave order parameters observed close to the transition are also reflected in the corresponding mean-field parameter.

## Acknowledgements

**Funding information**    S.K. acknowledges financial support from the Postdoctoral Fellowship from Institut Quantique, from UF Project No. P0324358 - Dirac postdoc fellowship, sponsored by the Florida State University National High Magnetic Field Laboratory (NHMFL) and from NSF DMR-2128556. D.S. acknowledges support by the Natural Sciences and Engineering Research Council of Canada (NSERC) under grant RGPIN-2020-05060. Computational resources were provided by the Digital Research Alliance of Canada and Calcul Québec.

## A   The inter-cluster mean-field procedure

The extended interaction term can be rewritten as

$$\frac{1}{2}\sum_{\mathbf{r},\mathbf{r}',\sigma,\sigma'}V_{\mathbf{rr}'}n_{\mathbf{r}\sigma}n_{\mathbf{r}'\sigma'} = \frac{1}{2}\sum_{\mathbf{r},\mathbf{r}',\sigma,\sigma'}V_{\mathbf{rr}'}^{\mathrm{c}}n_{\mathbf{r}\sigma}n_{\mathbf{r}'\sigma'} + \frac{1}{2}\sum_{\mathbf{r},\mathbf{r}',\sigma,\sigma'}V_{\mathbf{rr}'}^{\mathrm{ic}}n_{\mathbf{r}\sigma}n_{\mathbf{r}'\sigma'},$$

where $\mathbf{r},\mathbf{r}'$ refer to the lattice sites, and $n_{\mathbf{r}\sigma}$ is the number of particles at site $\mathbf{r}$ with spin $\sigma$. Here $V_{\mathbf{rr}'}^{\mathrm{c}}$ and $V_{\mathbf{rr}'}^{\mathrm{ic}}$ refer to the intra-cluster and inter-cluster parts of the interaction. Inspired by Wick's theorem, we decompose the inter-cluster part of the interaction into Hartree, Fock

and anomalous channels, as follows:

$$
\begin{aligned}
\frac{1}{2}\sum_{\mathbf{r},\mathbf{r}',\sigma,\sigma'} V_{\mathbf{rr}'}^{\mathrm{ic}} n_{\mathbf{r}\sigma} n_{\mathbf{r}'\sigma'} =& \sum_{\mathbf{r},\mathbf{r}',\sigma,\sigma'} V_{\mathbf{rr}'}^{\mathrm{ic}}\left( n_{\mathbf{r}\sigma}\bar{n}_{\mathbf{r}'\sigma'} - \frac{1}{2}\bar{n}_{\mathbf{r}\sigma}\bar{n}_{\mathbf{r}'\sigma'}\right) \\
& - \sum_{\mathbf{r},\mathbf{r}',\sigma,\sigma'} V_{\mathbf{rr}'}^{\mathrm{ic}}\left( f_{\mathbf{rr}'\sigma\sigma'}\bar{f}_{\mathbf{rr}'\sigma\sigma'}^* - \frac{1}{2}\bar{f}_{\mathbf{rr}'\sigma\sigma'}^*\bar{f}_{\mathbf{rr}'\sigma\sigma'}\right) \\
& + \frac{1}{2}\sum_{\mathbf{r},\mathbf{r}',\sigma,\sigma'} V_{\mathbf{rr}'}^{\mathrm{ic}}\left( \Delta_{\mathbf{rr}'\sigma\sigma'}\bar{\Delta}_{\mathbf{rr}'\sigma\sigma'}^* + \Delta_{\mathbf{rr}'\sigma\sigma'}^\dagger\bar{\Delta}_{\mathbf{rr}'\sigma\sigma'} - \bar{\Delta}_{\mathbf{rr}'\sigma\sigma'}\bar{\Delta}_{\mathbf{rr}'\sigma\sigma'}^*\right),
\end{aligned}
\tag{A.1}
$$

where the operators are defined as $n_{\mathbf{r}\sigma} \equiv c_{\mathbf{r}\sigma}^\dagger c_{\mathbf{r}\sigma}$, $f_{\mathbf{rr}'\sigma\sigma'} \equiv c_{\mathbf{r}\sigma}^\dagger c_{\mathbf{r}'\sigma'}$ and $\Delta_{\mathbf{rr}'\sigma\sigma'} \equiv c_{\mathbf{r}\sigma} c_{\mathbf{r}'\sigma'}$. Note that the applicability of Wick's theorem is not exact in this case, as we are considering a model which already includes on-site interactions, but must be considered as an *ad hoc* Ansatz. In other words, at a fundamental level, we are not assuming that the ground state of the system is a Slater determinant. We are rather resting on a variational principle for the self-energy [124] on which CDMFT is formally based.

The sum over sites $\mathbf{r}, \mathbf{r}'$ is taken over the whole lattice. But the average $\bar{n}_{\mathbf{r}\sigma}$ will be assumed to have the periodicity of the cluster, i.e., $\bar{n}_{\mathbf{r}+\mathbf{R}\sigma} = \bar{n}_{\mathbf{r}\sigma}$ where $\mathbf{R}$ belongs to the super-lattice. In addition, the two-site averages $\bar{f}_{\mathbf{rr}'\sigma\sigma'}$ and $\bar{\Delta}_{\mathbf{rr}'\sigma\sigma'}$ are assumed to depend only on the relative position $\mathbf{r}-\mathbf{r}'$. The mean-field inter-cluster interaction (A.1) is then a one-body contribution to the Hamiltonian with the periodicity of the super-lattice, and contains both intra-cluster and inter-cluster terms, whereas the purely intra-cluster part $V_{\mathbf{rr}'}^{\mathrm{c}}$ retains its fully correlated character.

For a four-site cluster, we have a total of eight bonds between neighboring clusters, along the $x$ and $y$ directions, with two spin combinations $(\sigma, \sigma')$ per bond, where we consider spin-parallel combinations for the Fock terms (in the absence of spin-dependent hopping) and spin-antiparallel combinations for the anomalous terms. In practice, we only consider physically relevant combinations of operators defined on different sites/bonds for our analysis (such as those compatible with a $d$-wave or an extended $s$-wave order). As an illustration of this, let us consider the pairing fields $\Delta$ defined on all of these bonds, which we denote by the labels $i = 1-16$ (including different bond and spin combinations).

The mean-field Hamiltonian can be written as

$$
\frac{V}{2}\sum_{i,j}\left( \bar{\Delta}_i^* M_{ij}\Delta_j + \Delta_i^\dagger M_{ij}\bar{\Delta}_j - \bar{\Delta}_i^* M_{ij}\bar{\Delta}_j\right),
\tag{A.2}
$$

where $i, j = (\mathbf{r}, \mathbf{r}', \sigma, \sigma')$ and the matrix $M_{ij}$ describes the combinations of the pairing fields defined on different bonds which appear in the Hartree-Fock decomposition of the inter-cluster interactions. The matrix $M$ turns out to be an identity matrix for the Fock and pairing fields $f$ and $\Delta$ respectively, but the corresponding matrix for the Hartree fields $n$ is off-diagonal.

Defining the eigen-combinations of the pairing fields by

$$
E_\alpha = U_{\alpha i}\Delta_i,
\tag{A.3}
$$

and the eigenvalues of the matrix $M$ by $\lambda_\alpha$, such that

$$
M_{ij} = \sum_{\alpha,\beta} U_{\alpha i}^* \lambda_\alpha \delta_{\alpha\beta} U_{\beta j},
$$

we can rewrite Eq. (A.2), above, as

$$
\frac{V}{2}\sum_\alpha \lambda_\alpha \left( \bar{E}_\alpha^* E_\alpha + E_\alpha^\dagger \bar{E}_\alpha - \bar{E}_\alpha^* \bar{E}_\alpha\right).
\tag{A.4}
$$

The mean-field values $\bar{E}_\alpha$ of the relevant eigen-combinations $E_\alpha$ of the pairing operators defined on different nearest-neighbor bonds are obtained self-consistently within the CDMFT loop, and likewise for the other mean fields that are the appropriate eigen-combinations of $\bar{n}_{\mathbf{r}\sigma}$ and $\bar{f}_{\mathbf{r}\mathbf{r}'\sigma\sigma'}$.

## B    CDMFT convergence

The CDMFT procedure is iterative and aims at finding a solution to a set of nonlinear equations that can be schematically expressed as

$$\mathbf{x} = \mathbf{f}(\mathbf{x}), \tag{B.1}$$

where $\mathbf{x}$ stands for the set of bath and inter-cluster Hartree-Fock parameters and $\mathbf{f}$ is an equally large set of functions that returns the next set of parameters from the current set, following a procedure that combines the CDMFT update with the inter-cluster mean-field one. The canonical way to solve Eqs (B.1) is the fixed-point method: the map $\mathbf{x}_{n+1} = \mathbf{f}(\mathbf{x}_n)$ is iterated until the difference $\Delta\mathbf{x}_{n+1} = \mathbf{x}_{n+1} - \mathbf{x}_n$ is smaller than some preset accuracy.

However, if the purpose is to find a solution to (B.1), there are more efficient and stable alternatives. Specifically, one could use the classic Broyden method for finding roots of sets of nonlinear equations, a generalization to many variables of Newton's root-finding method. Broyden's method relies on a computation of the Jacobian matrix $\mathbf{J} = \partial\mathbf{f}/\partial\mathbf{x}$ that is improved at each iteration. It typically finds a solution with fewer iterations than the fixed-point method, and with greater accuracy. In addition, it is "stickier", meaning that upon performing an external loop over model parameters, it will "stick" to the current solution (or the current phase), whereas the fixed-point method will be prone to instabilities and will more likely switch to more stable solutions.

This means that the fixed-point method, although less efficient, is more appropriate to detect phase transitions, whereas the Broyden method is better at keeping the current solution into its metastable regime. Hence the Broyden method will typically result in wider hysteresis loops than the fixed-point method when the external parameter is cycled in both directions (ascending and descending).

In practice, we can converge the CDMFT-DHF procedure on the difference $\Delta\mathbf{x}_{n+1}$, but we can also ask for the convergence of physical quantities, such as the cluster self-energy $\Sigma(\omega)$, or relevant order parameters. It may happen that physical quantities converge even though bath parameters do not, because the latter are sometimes subject to discrete "gauge" symmetries that do not affect physical observables. But even though convergence criteria may be based on physical quantities, the iteration $\mathbf{x}_n \rightarrow \mathbf{x}_{n+1}$ is still based on either the fixed point or the Broyden method. In this work, we used the self-energy and relevant order parameters as convergence criteria, with accuracies of the order of $10^{-4}$.

As an illustration, we compare the behavior of the relevant order parameters for the phases observed at the particle-hole symmetric chemical potential as a function of $U > 0$ for $V = -0.6$ in Fig. 15 and as a function of $V < 0$ for $U = 2$ in Fig. 16, using the fixed-point and the Broyden methods for obtaining the optimal set of CDMFT parameters. As expected, the region of hysteresis is found to be much larger when the Broyden method is used, consistent with the tendency of this method to stick to the current solution. Interestingly, we find that the existence of $d$-wave order does not necessarily coincide with phase separation, and there may be a region with a nontrivial $d-$ order parameter even at half-filling. However, such a region is not easily observed with the fixed-point method and is usually significantly amplified when the Broyden method is used, as illustrated in the lower plot of Fig. 15. We also observe oscillations between the $d$-wave solutions obtained in the presence and absence of phase separation within

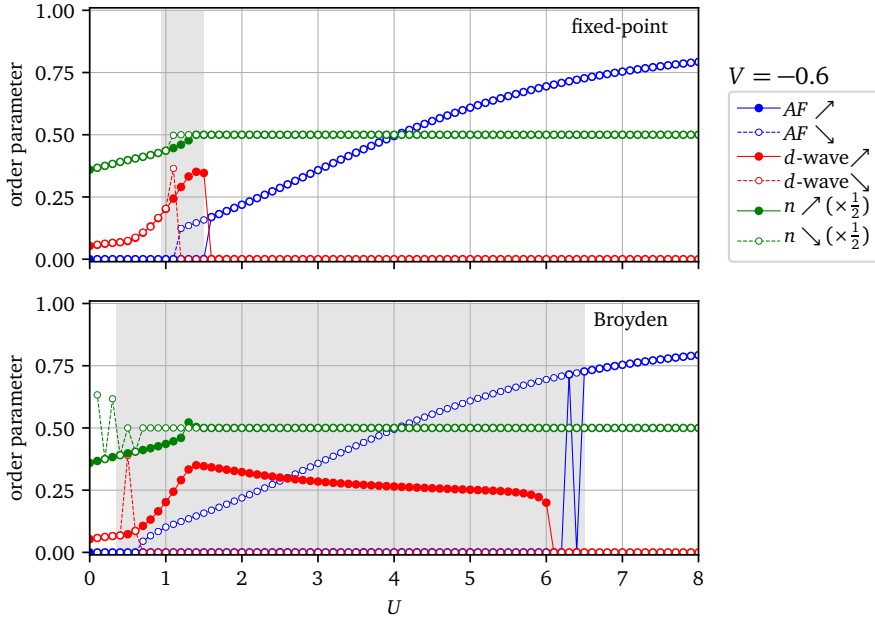

Figure 15: Order parameters for the different phases observed at the particle-hole symmetric chemical potential for $V = -0.6$, as a function of $U$, using the fixed-point method (above) and the Broyden method (below) for obtaining the optimal set of bath and mean-field parameters. The hysteresis loop obtained for increasing and decreasing $U$ is found to be much larger for the Broyden method, indicating that it has a tendency to stick to the current solution. A prominent region with a nontrivial $d$-wave superconducting order parameter is observed at half-filling for the Broyden method (indicated by the region with filled red circles in the lower plot). The transition from the phase-separated to the half-filled state is indicated by a shoulder-like feature in the corresponding $d$-wave order parameter. Oscillations are observed between the $d$-wave solutions with and without phase separation, within the hysteresis region, for both methods. In the presence of phase separation, the density is found to oscillate between values greater than and less than 1, when the Broyden method is used, and sometimes also with the fixed-point method. Moreover, some oscillations are also observed between the AF and normal states, close to the phase transition towards AF for increasing $U$ (see open blue circles in the lower plot).

the gray hysteresis region, for both the methods. Although the Broyden method converges faster even with a higher accuracy, we obtain more oscillatory solutions in general with this method, which includes oscillations between densities greater than and less than 1 in the phase-separated region for small $U$, as well as between the normal state and the AF state, close to the transition from $d$-wave to antiferromagnetism for increasing $U$. In Fig. 16, we see that $d$-wave superconducting state persists well into the region of half-filling as $V$ becomes less negative, for both methods. When the Broyden method is used, we find that the system continues in the AF state down to $V = -1.8$ and then undergoes a transition to the normal state, without the appearance of a $d$-wave order or phase separation. This is an extreme example of the tendency of this method to preserve the existing solution. In contrast, the fixed-point method gives rise to a phase transition towards the $d$-wave superconducting state, close to V=-0.9. Therefore, for most situations, it is more convenient for us to employ the fixed-point method for our computations.

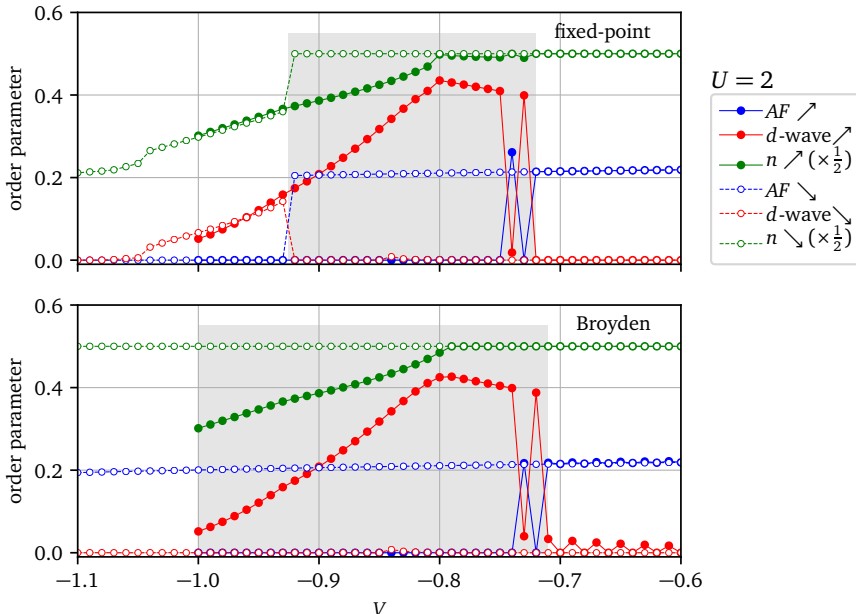

Figure 16: Order parameters corresponding to the different phases observed at the particle-hole symmetric chemical potential for $U = 2$, as a function of $V$, using the fixed-point method (above) and the Broyden method (below) for obtaining the optimal bath and mean-field parameters. Once again, the hysteresis region between increasing and decreasing negative $V$ is found to be much larger when the Broyden method is employed. Interestingly, the AF region is found to persist all the way to $V = -1.8$ for decreasing (more negative) $V$ with the Broyden method (not shown in the figure), beyond which the system directly undergoes a transition to the normal state, and the intervening $d$-wave superconducting region is found to be absent (the open blue circles in the lower plot depict the behavior up till $V = -1.1$). For increasing (less negative) $V$, a part of the $d$-wave superconducting phase observed is found to be very close to half-filling for both methods (indicated by the filled red circles). Moreover, oscillations are observed between the $d$-wave and AF phases, which are found to occur more frequently when the Broyden method is used. Note that the results for increasing $V$ have been plotted starting from $V = -1.0$ in both cases for convenience, but may be smoothly extrapolated to more negative values of $V$.

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
