# Peer review of "CDMFT+HFD : an extension of dynamical mean field theory for nonlocal interactions applied to the single band extended Hubbard model"

_SciPost Physics Core, doi:SciPost Phys. Core 7, 033 (2024)_

## Round 1 · Referee Report · Anonymous (Referee 1) · 2024-1-4

Strengths
1- The authors provide a detailed and well-cited presentation of the different studies previously performed on the Extended Hubbard Model (EHM).
2- The manuscript is written in a clear way and it leaves no ambiguities.
3- This work illustrates the significance of the non-local (inter-cluster) interactions by comparing the results of CDMFT+HFD to those of simple CDMFT. It is shown that certain regions of superconducting order appear only upon inclusion of the inter-cluster interactions.
4- The method appears to be agile with no significant restriction in its applicability and can be used in the future for studies of more complicated systems (e.g. the authors suggest the study of systems with spin-orbit interactions).
Weaknesses
1- The results are in good qualitative agreement with previous works, however there is not significant originality in the findings.
2- This approach is not suitable for studying incommensurate magnetic orders, which are found in the literature to be realized at certain regimes of the phase diagram.
Report
The authors study the competition between different phases for a repulsive local Coulomb interaction ($U>0$) and for both a repulsive and an attractive nearest-neighbor interaction ($V>0$ and $V<0$ respectively). They work on two different impurity models, a more general one and a simplified one, and they compare the relevant results. Different regimes of the phase diagram are investigated (with respect to interaction strength and doping) and both spin/charge ordered phases as well as superconducting phases are investigated.
The authors find that for repulsive nearest-neighbor interactions and strong local repulsion a first-order transition from an antiferromagnetic to a charge density wave state is observed. Upon hole-doping a $d$-wave superconducting dome is observed. For attractive nearest-neighbor interactions, at the particle-hole symmetric chemical potential, a first-order transition takes place from antiferromagnetism to $d$-wave superconductivity, accompanied by phase separation in the vicinity of half-filling. Away from half-filling, regions of $d_{x^2-y^2}$-wave superconducting order are found as well as disconnected pockets of extended $s$-wave order.
The results are in good qualitative agreement with previous works using other approaches. The CDMFT+HFD method is novel and it has the advantage of combining a non-perturbative treatment of local and short-range correlations with the inclusion of spatial fluctuations, in a mean-field manner. As the authors state in the Introduction "This work constitutes a test of the CDMFT+HFD method". To this end, even though there is no significant originality in the obtained results, since there is originality in the method, I believe the goal of the paper has been reached. In my view, the manuscript meets the criteria for publication in SciPost Physics Core and its publication would be beneficial to the community, as it validates the use of this new technique (I would be glad, however, to see the points raised in the "Requested changes" section addressed).
Requested changes
1- I believe it would be beneficial if the authors made a comment on the effect of the cluster size on the validity of the results. They only discuss a 4-site cluster, but they do not state if there is a significant technical limitation that prevents the increase of the cluster size. Given for example the observed jumps in the number density $n$ close to quarter-filling, at the attractive $V$ case, that are not well understood, I believe such a discussion might be enlightening.
2- It appears that the comparison between the behavior of the simple and general model as a function of density for $V<0$ (Fig. 7 and Fig. 8) suggests that the simple model can, in certain cases, give misleading results (see $s$-wave patches at $U>0$). Maybe the authors could make a comment on this conclusion, since it appears to be an important remark to keep in mind for future applications of the method.
-
Regarding the issue of cluster and bath size: The cluster and bath size are limited by the exact diagonalization solver: the upper limit for the total number of cluster and bath orbitals that one can realistically consider is 4+8=12, given that the ground state and Green function must be computed repeatedly in this approach. Of course, the same remark applies to single-site DMFT, with a cluster of size of 1! It is therefore impossible to perform a meaningful finite-size scaling analysis in this approach. The assumption is that any changes observed for larger system sizes would be quantitative in nature and that the basic dependence on model parameters or filling would be the same. We comment on this issue in the revised version. An additional remark is that we do not believe that the jumps observed in the number density close to quarter-filling are a result of the cluster size alone, as no such jumps are observed for the simple bath model even with the same cluster size.
-
Regarding the comparison between the simple and general bath models: The simple model provides a submanifold of the parameter space of the general model. Indeed, we find that this simple model can sometimes give rise to additional phases and the only way to verify the correctness of these results is to confirm them by using the general model. Unfortunately, we do not have a systematic understanding of why such differences appear. However, the results are for the most part qualitatively similar for both models, and we believe that the differences observed in the extended s-wave order for U>0 serve to provide an illustration of the advantages of considering a larger number of bath parameters in the CDMFT procedure. Indeed, this is a reason for testing two different bath models: to check the robustness of the results, and by and large the results are robust, with the exceptions pointed out by the referee.

Author: Sarbajaya Kundu on 2024-04-17 [id 4428]
(in reply to Report 2 on 2024-02-05)We thank the referee for this extensive and stimulating report.
Concerning the number of independent bath parameters. We agree that the statement about the number of bath parameters in the general model can be confusing for the reader without a more explicit explanation. We have added a sentence in the caption of Fig. 2, as well as in lines 177-178 of the main text, explaining that for the general model, we have three bath parameters per bath orbital, which leads to 24 parameters in total, and then we subtract 6 of those due to the presence of a C4v rotational symmetry, giving us 18 bath independent parameters in this case.
Concerning the symmetry of the phase diagram with respect to half-filling. Indeed, the phase diagram must be symmetric about half-filling. Based on our simulations, we find that the observed asymmetry in the s-wave region present near the extreme band edges in the figure for U=0,V=-0.4 is a consequence of insufficient accuracy in the CDMFT computations, as opposed to any new physics. We now comment on this observation in the caption of Fig.7. We have also now replaced Fig.7 of the main text with results obtained with a higher accuracy for the CDMFT procedure, wherever possible. However, we now find some asymmetry between the d-wave orders on both sides of half-filling for U=0,V=-0.4, which we clearly identify to be the result of lack of convergence and consequently, a lower accuracy being used, in that particular region. This being said, this symmetry may be broken depending on the sweep direction of the chemical potential, but is restored when we sweep in the opposite direction, with possible hysteresis, hence the necessity of covering the full range from n=0 to n=2. This also serves as a sanity check on the accuracy of the method.
Regarding the presence or not of the pairing mean-field E_s. While it is true that there are some additional points obtained in the range 0.6<n<0.65 that are found only in the absence of the extended s-wave anomalous mean-field parameters E_s, the values of the s-wave order parameter at these points are found to be below the accuracy of the CDMFT procedure for these computations; hence we do not comment on these differences in the paper. We agree with the referee that the strongest mismatch between the calculations with and without E_s is found for V=-0.7 in the density range 0<n<0.3; we have now added a remark about this in the caption of Fig.13.
On the issue of whether one can say that the exact result would lie somewhere between the C-DMFT and C-DMFT + mean-field results. This is an interesting point. We believe it is too simplistic to say that the exact ground state properties lie in between the two descriptions, since increasing the cluster size might further suppress the ordering tendency due to longer-range fluctuations within the cluster. Hence, we are unable to take a firm stand on this point.
Regarding the eventual disappearing of first-order transitions upon increasing the cluster size. In general, we do expect that increasing the cluster size will lead to smaller hysteresis regions because of enhanced fluctuations. However, it is impossible to say a priori whether these effects will turn first order transitions into continuous transitions. After all, first-order transitions between states that share the same symmetry do exist in nature. Dimensionality is probably an important factor. Our results still have a mean-field character; the effect of increasing the cluster size within a purely two-dimensional model (more fluctuations) might be compensated by adding a weak hopping term in the third dimension (less fluctuations), when considering a three-dimensional model.
Regarding an effective attraction coming from non-perturbative scattering processes at strong coupling. We agree that it is important to discuss the change in the behavior of the phase separation when we consider a stronger on-site repulsion, and it is possible that an analogous situation does arise at strong coupling, even though there is no evidence for continuity between the weak- and strong-coupling limits. Nevertheless, it would be interesting to explore the physics in the strong-coupling limit using our method and we have commented on this in the revised version.
About extending the conclusion. We thank the referee for these valuable suggestions, which not only provide us with new perspectives for possible applications of our method, but also contribute to the usefulness of the paper for readers. We have added the suggested remarks to the final part of the conclusion.
We have now replaced the word “pockets” in the abstract and in some other parts of the paper with ”regions/islands”.

---

## Round 1 · Referee Report · Anonymous (Referee 2) · 2024-2-5

Report
Their investigation has been carried out by means of a reliable many-body approach, the Cluster (or Cellular)-dynamical mean-field theory (C-DMFT), which captures non-perturbatively the electronic correlations within the clusters considered and which has been further supplemented by a conventional mean-field treatment of the inter-cluster effects.
The results presented in the manuscript, while essentially confirming some of the qualitative trends emerged in previous, more sporadic investigations of the literature, allow the Authors to outline a quite general picture of the physics driven by non-local electronic interactions in two-dimensional lattice systems.
In particular, I have appreciated the well conceived organization of the presentation of the data and of the associated physical discussions, and the interrelation the Authors make between the different regimes considered. This renders the reading of the whole manuscript quite pleasant and instructive.
On the basis of these considerations, I recommend the publication of the manuscript by Kundu and Senechal on SciPost Physics-Core, after the Authors have (optionally) considered the specific observations I am including below, aimed at further improving the clarity and the impact of their paper.
Requested changes
Optional observations/suggestions for changes:
1) In the presentation of the method, I find a bit confusing the description of the number of (bath) parameters. In particular, while the fact that they should be 6 for the simple impurity case, illustrated in Fig. 1 is quite evident, that their number rises for the general impurity model up to 18 is not immediately clear understandable by looking at Fig. 2, as well as from the related discussion at p. 6, lines 165-171. Maybe the Authors could extend/make the connections between the Figure and manuscript more transparent on that specific point.
2) As the model considered has no frustration, I would have expected a completely symmetric shape of the different phases appearing in Fig. 7, as a function of filling (with the possible exception of first order coexistence features). This expectation seems indeed to be met except for the small s-wave region present at the extreme band edge (n=0) but not on the opposite side (n=2) in the first panel of the Figure (U=0, V= -0.4) or displaying an apparently different shape in the fourth panel (U=2, V= -0.7). Is this just a numerical effect, or is such generic expectation not fully correct?
3) While the overall trend of suppressing/enhancing the long-range ordered phases by excluding/including the (corresponding) self-consistent mean-field parameters (E_s, E_d, etc ...) is clear, the data shown in Fig. 13 seem to display also some exception, such as for the s-wave order without E_s, which appears between 0.6 <n < 0.65. Further, it looks to me, that although the strongest mismatch between the calculations w/wo E_s is found for the case of V=-0.7 in the density range of 0 <n < 0.3, this is not explicitly remarked in the Caption as well as in the manuscript text.
4) Somewhat related to this: The Authors correctly mention in their discussions, that the inclusion of intra-cluster correlation reduces the ordering tendency of the system w.r.t. the pure mean-field, while they find (which is quite reasonable) that the intercluster mean-field self-consistency partially "re-enhances back" the ordering tendency. On this basis, would it be reasonable, then, to expect that the (unknown) exact ground state properties lie somewhat in between those two description (C-DMFT) and (C-DMFT + mean field) ? Or the limitation to small cluster size would prevent to draw such conclusion?
5) In the calculations based on diagrammatic extensions of DMFT (such as the Dynamical Vertex Approximations, the Dual Fermion/Dual Boson approach, etc.), it was shown that for the unfrustrated 2D Hubbard model for V=0 the metal-insulator transitions described by the DMFT MIT was shifted down to U -> 0 for T-> 0, with some hints that its 1st order nature might be dissolved in a sharp crossover at finite T. Would the Authors expect a similar trend might affect their 1st order transitions, if considering correlation on larger cluster size?
6) At the end of p. 12 the Authors state that the region of phase-separation shrinks with increasing on-site repulsion U. However, while this is intuitively understandable in the weak-coupling limit, there might be exceptions at stronger coupling. I refer here to the finite temperature analysis of , e.g., PRB 99, 035161 (2019) and PRL 125, 196403 (2020), where it was shown how nonperturbative scattering processes arising from the strong on-site repulsion might appear, to a certain extent, as an effective attraction. An extension of the discussion on this point might be then considered (see also below).
7) In the interest of the manuscript impact, one could consider to extend the final part of the conclusion, by presenting more concrete outlooks for this work. For instance, would it be technically possible to perform such systematic investigation for single orbital Hubbard model on triangular lattices? If yes, this might worth to be mentioned, as the importance of non-local interactions (even long-range ones!) have been pointed out in the literature (see e.g., PRL 110, 166401 (2013). Further, in the light of 6), it could be interesting to extend, in the future, the Authors' calculations to the case with V<0 and larger U values to estimate their intertwined effect in the nonperturbative regime in driving/suppressing the tendency to phase-separation. Finally, one could mention the insertion of a geometrical frustration (eg, via a next to nearest hopping term), which would be also important for make a more precise connection to the physics of cuprates, in the case this might be incorporated within the ED implementation (for the four-site cluster) adopted by the Authors.
8) Eventually, I would replace the word "pockets" in the abstract with some equivalent one, to avoid confusion with the usual meaning (referred to the Fermi Surface) with which this term is used to describe the physics of cuprates.

---

## Round 2 · Referee Report · Anonymous (Referee 1) · 2024-5-1

Report

The authors have improved consistently the manuscript, incorporating the suggestions of the two referees. Moreover, new calculations provide results with a higher accuracy and the addition of a new approach (the Broyden method) to converge the CDMFT parameters leads to a detailed, instructive analysis on the technical aspects of the CDMFT method.
I believe that the manuscript at its current version meets the expectations and criteria of SciPost Physics Core and it warrants publication in the journal.

Recommendation

Publish (easily meets expectations and criteria for this Journal; among top 50%)

---

## Round 2 · Referee Report · Anonymous (Referee 2) · 2024-5-4

Report

The Authors have made a very good job in answering the questions posed the Referees. The presentation of their results and the precision of the statements made in the resubmitted manuscript have definitely improved, which will likely further enhance the paper's impact. Hence, I recommend the revised version of the manuscript by Sarbajaya Kundu and David Senechal to be published in SciPost Physics Core.

Recommendation

Publish (easily meets expectations and criteria for this Journal; among top 50%)

---

## Round 2 · Author Response

Dear Editor,

We would like to resubmit our manuscript, titled “CDMFT+HFD: an extension for dynamical mean field theory for non-local interactions applied to the single-band extended Hubbard model”. We regret the delay in the resubmission, largely owing to the performance of some new computations.
We have responded to the questions and remarks of the referees in the space provided, and are attaching here a copy of the revised manuscript, followed by another copy of it highlighting the changes with respect to the last submitted version. Below, we list the changes made.
Apart from incorporating the suggestions of the referees, we have replaced figures 3, 4 and 7 in the manuscript by more accurate but qualitatively similar versions, and have modified the corresponding captions to include discussions on the new features in these figures. We have also performed new computations using a different method to converge the CDMFT parameters, which we chose not to use for the figures in the main text due to certain limitations, but have described in a new appendix (Appendix-B). We have also added new figures 15 and 16 to the paper, illustrating the differences between the existing fixed-point and the new Broyden method.

Thanking you,
Sincerely,
Authors.

---

## Round 2 · List of Changes

Let us summarize the main changes brought to the manuscript since the first version.

Changes suggested by the referees:

  1. We have commented on the difficulty in finite-size scaling in our CDMFT approach in lines 153-159 of the manuscript.

  2. We have added an explanation for counting the number of bath parameters for the case of the general model in the caption of Fig.2, as well as in lines 177-178.

  3. The asymmetry between the orders on either side of half-filling, particularly for U=0,V=-0.4, has been addressed in the caption of Fig.7.

  4. In lines 286-289, we have added a remark about the advantage of considering a general bath model along with the simple bath model, as evidenced by the change in the behaviour of the extended s-wave order as a function of filling, for increasing U.

  5. In the caption of Fig. 13, we have added a remark about the prominence of the difference with and without the anomalous mean-field parameters for density in the range 0<n<0.3.

  6. We have added new remarks in the final paragraph of the conclusion, in lines 398-404, about the application of our method to the single-band Hubbard model on a triangular lattice, the possibility of exploring the regime of non-perturbative repulsive local interactions and attractive non-local interactions and of including longer-range hopping terms.

  7. We have replaced the term “pockets” with “regions” or “islands” everywhere it occurs in the manuscript.

Additional changes:

  1. In lines 182-184, the number of bath parameters at half-filling for the general model has been changed to reflect the effectively reduced set of parameters due to the imposition of particle-hole and four-fold rotation symmetries, as opposed to the actual (larger) number used in the computations, which were mentioned in the previously submitted version. In lines 170-172, the number of bath parameters at half-filling for V<0 for the simple model increases from 6 to 10 because particle-hole symmetry is no longer imposed.

  2. We have replaced figures 3, 4 and 7 in the manuscript. These have been treated with a greater accuracy and for the computations at half-filling for V<0, where phase separation occurs, the imposition of particle-hole symmetry on the bath parameters has been removed as it was in contradiction with the lack of restrictions on the corresponding mean-field parameters.The captions have been modified to include observations about the new figures, which are qualitatively similar. Correspondingly, figures 9 and 14, derived from the same data sets, have been modified for consistency.

  3. We performed new computations using a different method (the Broyden method) to converge the CDMFT parameters. This is described in a new appendix (Appendix B) of the paper, and figures 15 and 16 have been added, which illustrate the differences between the results obtained using the existing fixed-point method and the Broyden method for V=-0.6 as a function of U and U=2 as a function of V<0, respectively, at half-filling. This, and point 2 above, explain the delay between the first referee reports and this new version.

---

## Editorial Decision

published